# Gender dynamics in the relationship among student-teacher relationships, academic task engagement, academic task value, and positive developmental outcomes

Getahun Tadesse Abren*, Reda Darge Negasi, Amare Sahle Abebe

Department of Psychology, School of Educational Sciences, College of Education, Bahir Dar University, Bahir Dar, Ethiopia

These authors equally contributed to this work.
* gechpsycho03@gmail.com

## Abstract

This study examines gender differences in the direct and indirect relationships among student-teacher relationships, academic task engagement, perceived academic task value, and positive developmental outcomes among secondary school students. A correlational research design was used, and a cross-sectional survey was conducted with 500 students from grades 9 and 10 across five selected secondary schools. Multi-stage sampling technique employed to select participants. The study utilized Multi-Group Structural Equation Modeling (MGSEM) to examine gender difference. The results indicate that the student-teacher relationship and students' perceived academic task value were associated with 40.3% of the variance in cognitive engagement among male students ($R^2 = 0.403$), while they explain 31.9% of the variance for female students ($R^2 = 0.319$). Regarding affective engagement, these factors were associated with 15.0% of the variance for males ($R^2 = 0.150$) and 12.6% for females ($R^2 = 0.126$). In terms of behavioral engagement, the model explains 26.4% of the variance for males ($R^2 = 0.264$) and 32.8% for females ($R^2 = 0.328$). Furthermore, the model were associated with 52.3% of the variance in positive development outcomes for male ($R^2 = 0.523$) and 54.9% for female ($R^2 = 0.549$). These findings underscore the necessity for gender-sensitive educational strategies that strengthen student-teacher relationships and tailor engagement approaches to promote positive developmental outcomes for both genders. Specifically, educators should focus on fostering supportive environments that recognize the unique needs of male and female students, thereby promoting higher levels of academic task engagement and positive developmental outcomes.

**Data availability statement:** All relevant data are within the manuscript and its Supporting Information files.

**Funding:** The author(s) received no specific funding for this work.

**Competing interests:** The authors have declared that no competing interests exist.

## Introduction

In educational settings, boys and girls often exhibit different levels of motivation, engagement, and classroom behavior. For instance, research by [1] found that girls generally report higher academic motivation, while boys' motivation is more strongly associated with their classroom behavior. The study also revealed that different aspects of motivation (e.g., cognitive for boys and behavioral for girls) are predictive of their respective behaviors in class. Societal gender expectations and cultural norms can shape students' academic self-concept and perceptions of task value, thereby influencing their motivation, learning strategies, and academic performance [2]. These influences may also extend to career aspirations, as research suggests that gendered beliefs about ability and value affect students' educational choices and long-term goals [3,4].

Building on this understanding, the student-teacher relationship is widely recognized as a foundational element shaping students' academic experiences and developmental trajectories. Research consistently demonstrates that high-quality student-teacher relationships foster greater academic engagement, intrinsic motivation, and achievement [5,6]. However, existing study findings suggest that these relationships are not experienced uniformly across genders. For instance [7] found that girls often flourish in relationally supportive and collaborative environments, whereas boys respond more favorably to structured and performance-oriented interactions.

These gender-based preferences also extend to academic task engagement, which refers to students' active participation of emotional, behavioral, and cognitive investment in learning tasks [8]. Regarding gender differences, research suggests that girls often exhibit higher levels of cognitive and emotional engagement, particularly in cooperative and relationally supportive learning contexts [9,10]. In contrast, boys showed more engagement in competitive or highly structured activities but tended to display lower sustained cognitive engagement in traditional classroom settings [7].

Closely linked to academic task engagement is academic task value, defined as students' perception of a task's importance, usefulness, and interest of academic tasks, which are critical for sustaining motivation [11]. From the perspective of Expectancy–Value Theory [11,12] students' motivation and achievement-related behaviors are shaped by both their expectations of success and the value they attach to academic tasks. Tasks perceived as valuable, because they are useful, important, or enjoyable encourage sustained cognitive, behavioral, and emotional engagement. This sustained engagement can, in turn, promote positive developmental outcomes, including the Five Cs of competence, confidence, connection, character, and caring [13]. Gender disparities have been observed, with girls often valuing tasks that align with personal relevance and long-term goals, while boys may prioritize tasks that offer immediate utility or a competitive edge [2]. These differences in perceived value significantly influence patterns of engagement and academic achievement across subject areas.

Such differences in student-teacher relationships, academic task engagement, and perceived value of academic tasks might contribute to variations in students' positive developmental outcomes. Positive developmental outcomes including confidence,

competence, connection, character, and caring, are central indicators of holistic development [14–16]. Research reveals persistent gender patterns across these domains, with girls often excelling in social-emotional competencies such as connection and caring, while boys tend to demonstrate strengths in confidence and leadership, particularly in competitive or performance-based contexts [17,18]. These differences are further shaped by cultural norms surrounding gender roles, which influence students' perceptions of academic relevance and their interactions with teachers [10]. In addition, teacher expectations were found to be frequently influenced by implicit gender biases, which in turn, have significantly affected students' academic motivation and achievement [19]. Without adequate attention to these gendered dynamics, educational systems risk reinforcing existing inequities, especially in contexts where systemic gender disparities are already present.

Despite a growing body of research conducted worldwide, there remains an inadequacy of studies employing a comprehensive analytic framework, such as Multi-Group Structural Equation Modeling (MGSEM), to extricate the gender-specific pathways linking student-teacher relationships, academic task engagement, perceived academic task value, and positive development outcomes, especially in African contexts such as Ethiopia. Addressing this gap is critical for informing gender-sensitive educational practices that promote equitable developmental opportunities for all students.

To address this gap, this study explored gender differences in the interrelationships among student-teacher relationships, academic task engagement, perceived academic task value, and positive development outcomes, offering an understanding necessary for advancing inclusive education policies and practices. Specifically, the study addresses the research question of: *How gender moderates the relationships among student–teacher relationship, academic task value, dimensions of academic task engagement, and positive developmental outcomes among secondary school students?*

## Method and materials

### Research design

This study aimed to explore gender dynamics in the relationship among student – teacher relationship, academic task engagement, academic task value, and students' positive development outcomes. A correlational research design was employed for this investigation. This design was selected because it facilitates the examination of the degree of association among multiple variables at a single point in time, thereby enabling researchers to understand how variations in independent variables correspond to changes in dependent variables [20].

### Participants

The participants in this study were Grade 9 and 10 students from both public and private secondary schools within the Bahir Dar city administration, which includes a total of 21 secondary schools (i.e., 11 public and 10 private) with a combined enrollment of 16,725 students in the 2024 academic year (14,690 in public and 2,035 in private schools). A multi-stage sampling technique was employed to ensure representativeness and methodological rigor. In the first stage, five schools (three public and two private) were randomly selected. In the second stage, 13 classrooms from Grades 9 and 10 were randomly selected using a lottery method within the chosen schools. Initially, 500 students were recruited to participate in the study. Of these, 205 (41%) were male and 295 (59%) were female. The age of the participants ranged from 14 to 22 years, with an average age of 16.69 years. The majority of students (404 or 80.8%) were enrolled in public schools, while 96 (19.2%) attended private schools. After excluding 17 incomplete responses, the final sample for analysis comprised 483 students, including 197 males (40.8%) and 286 females (59.2%), yielding a response rate of 96.6%.

### Measures

To ensure cultural and linguistic appropriateness, the researchers adapted instruments from established studies and implemented a rigorous two-step translation process. First, language experts translated the questionnaire into Amharic,

followed by a back-translation into English to verify accuracy. Expert reviewers considered local cultural norms and values throughout this process to maintain both linguistic integrity and cultural relevance. Additionally, the adapted instrument underwent expert review and pilot testing with a small group of students to identify and address potential misunderstandings or culturally inappropriate items. These procedures enhanced the reliability and validity of the data, contributing to a more accurate understanding of student – teacher relationship, perceived academic task value, academic task engagement and positive development outcomes within the Bahir Dar context.

**Positive youth development (PYD) five Cs scale – short form (PYD-5Cs SF – 34).** The researchers used the positive youth development (PYD) five Cs scale – short form (PYD-5Cs SF – 34) to collect data on students' positive development [21]. This self-report instrument originally included 34 items focused on the 5Cs: competence, confidence, character, connection, and caring. In the original study, the researchers found Cronbach's alpha (α) values greater than 0.80, indicating strong reliability. In the current study, the researchers calculated the internal consistency of the overall instrument's Cronbach's alpha (α) as 0.94, with sub-scale consistencies of 0.85 for confidence, 0.92 for competence, 0.90 for character, 0.90 for connection, and 0.84 for caring. The researchers also computed Confirmatory Factor Analysis (CFA), and the model fit indices showed the following: Comparative Fit Index (CFI) = 0.926, Tucker-Lewis Index (TLI) = 0.919, Chi-Square Minimum Discrepancy divided by Degrees of Freedom (CMIN/DF) = 1.539, and Root Mean Square Error of Approximation (RMSEA) = 0.052. Therefore, the model fits well.

**Competence**: The researchers measured competence with six (6) items covering academic, social, and physical self-perceived competence. Participants rated their agreement on a five-point Likert scale, ranging from 1 (Strongly Disagree) to 5 (Strongly Agree). Sample items included "I feel I am just as smart as others my age" (academic), "I have a lot of friends" (social), and "I think I can do well at just about any new athletic activity" (physical).

**Confidence**: The researchers measured confidence with six (6) items focusing on self-worth, physical appearance, and positive identity. Participants rated their agreement on a five-point Likert scale, ranging from 1 (Strongly Disagree) to 5 (Strongly Agree). Sample items included "I think I am good looking" and "All in all, I am glad I am me."

**Connection**: The researchers assessed connection with eight (8) items that addressed the youth's relationships with family, peers, school, and neighborhood. Participants responded on a five-point Likert scale, ranging from 1 (Strongly Disagree) to 5 (Strongly Agree) for family, school, and neighborhood items, and from 1 (Almost Never True) to 5 (Always True) for peer items. Sample items included "I get a lot of encouragement at my school" (school) and "My friends care about me" (peer).

**Character**: Character was measured with eight (8) items covering personal values and social consciousness. Participants rated their agreement on a five-point Likert scale, ranging from 1 (Not Important) to 5 (Extremely Important). Sample items included "Doing what I believe is right even if my friends make fun of me" (personal values) and "Giving time and money to make life better for other people" (social consciousness).

**Caring**: The researchers assessed caring with six (6) items focusing on empathy and sympathy. Participants rated how well each statement fit their perception of themselves on a five-point Likert scale, ranging from 1 (Not Well) to 5 (Very Well). Sample items included "When I see someone being taken advantage of, I want to help them" and "It makes me sad to see a person who doesn't have friends."

**Comprehensive task engagement questionnaire (Comp-TA questionnaire).** The researchers employed a self-reported questionnaire focused on engagement in academic tasks, referred to as the Comp-TA questionnaire. The reliability of the instrument was assessed, showing high internal consistency with a Cronbach's alpha (α) of 0.92. This questionnaire measures three dimensions: affective, cognitive, and behavioral [22].

Students rated their responses on a 5-point Likert scale (1 = totally disagree; 5 = totally agree). In the current study, the alpha coefficient (α) for the overall instrument was 0.88, with the following values for the sub-scales: 0.87 for cognitive engagement, 0.82 for behavioral engagement, and 0.79 for affective engagement. Confirmatory Factor Analysis (CFA)

was also conducted, and the model fit indices were (CFI = 0.936; TLI = 0.923; CMIN/DF = 1.891; and RMSEA = 0.066). These results suggest that the model has a good fit.

The instrument consists of 15 items distributed across the three dimensions of engagement in academic tasks. One, affective dimension – it includes four items (e.g., "When performing a task in class: They make me curious to learn new things"). Two, behavioral dimension – it includes seven items (e.g., "When performing a task in class: I follow the instructions given by the teacher for its development"). Third, cognitive dimension – it includes four items (e.g., "When performing a task in class: I think about what I already know about the topic because it can help me understand better").

**Student-teacher relationship measurement (STRM).** The study measured the student-teacher relationship using the adopted Student-Teacher Relationship Measure (STRM), which previous researchers [23] have used. This instrument consists of 25 items across two dimensions: social relationships and academic relationships. Researchers have recognized the STRM as a reliable and valid measure of the quality of student-teacher relationships. In the original study, Cronbach's alpha (α) coefficients were 0.92 for the academic relationship and 0.89 for the social relationship [23]. Similarly, we found high internal consistency reliability in the current sample, with a Cronbach's alpha (α) of 0.96 for the whole scale. Additionally, the Cronbach's alphas (α) values for the dimensions were strong, with a Cronbach's α of 0.95 for the academic relationship and 0.89 for the social relationship. Confirmatory Factor Analysis (CFA) was also conducted, and the model fit indices were (CFI = .916; TLI = .908; CMIN/DF = 2.017; and RMSEA = .071). These results suggest that the model has a good fit.

The academic relationship dimension includes 15 items, while the social relationship dimension includes 10 items. Examples of items from the academic relationship dimension are: "My teacher expects me to participate effectively in the classroom" and "My teacher shows remarkable enthusiasm while teaching the subject." Examples of items from the social relationship dimension are: "My teacher listens to what I say" and "My teacher links subject topics with characters that matter to us." Students rated their agreement with each statement using a 5 point Likert scale (1 = definitely does not apply, 2 = applies little, 3 = applies sometimes, 4 = applies often, and 5 = definitely applies).

**Students' perceived academic task value.** The study assessed students' academic task value using a self-reported subscale from the Motivated Strategies for Learning Questionnaire (MSLQ) [24]. This subscale contains six items that measure task value, focusing on students' perceptions of the importance, interest, and usefulness of academic tasks in their subjects. The original questionnaire had a Cronbach's alpha of 0.90 [21], while the current study showed an internal consistency Cronbach's alpha (α) of 0.82. An example item is: "I think I will be able to use what I learn in these courses in my future life." Students rated their agreement with each statement on a 5-point Likert scale (1 = not at all true of me, 2 = not true of me, 3 = undecided, 4 = true of me, and 5 = very true of me). Confirmatory Factor Analysis (CFA) was also conducted, and the model fit indices were (CFI = 0.998; TLI = 0.997; CMIN/DF = 1.059; and RMSEA = 0.017). These results suggest that the model has a good fit.

## Validations of the instruments

This study rigorously validated measurement constructs by analyzing content and construct validity, convergent and discriminant validity, and factor loading. Content and construct validity in this study was established through a systematic evaluation by six PhD candidates in Educational Psychology, ensuring that the measurement instruments accurately taken the intended constructs. This rigorous review led to critical feedback and the removal of inadequate items. Factor loading confirmed significant contributions of items to their respective constructs, while strong convergent validity was demonstrated with Average Variance Extracted (AVE) values exceeding the 0.50 threshold across all constructs. Discriminant validity was also supported, as shared variances between constructs remained lower than their AVE values, further enhancing the reliability of the research findings [25]. All instruments construct internal consistency Cronbach's alpha coefficient (α) values above 0.70 indicating acceptable reliability.

## Data collection

Data collection for this study took place from 25/10/2024–15/10/2024. A cross-sectional survey was conducted with 500 students from five selected secondary schools in Bahir Dar City Administration, Ethiopia.

## Methods of data analysis

To analyze the study's data, two statistical programs were employed: SPSS (Statistical Package for the Social Sciences) version 27 for descriptive statistics and AMOS (Analysis of Moment Structures) version 23 for confirmatory factor analysis (CFA) and multi-group structural equation modeling (MGSEM). MGSEM was utilized to examine gender differences in the relationships among student – teacher relationship, students' perceived academic task value, academic task engagement dimensions (cognitive, behavioral, and affective), and students' positive development outcomes. Four fit indices were used to evaluate model fit for all analyses. Specifically, Root Mean Square Error of Approximation (RMSEA), with acceptable fit defined as ≤ 0.08; Tucker-Lewis Index (TLI) and Comparative Fit Index (CFI), where acceptable fit is indicated by values ≥ 0.90; Chi-Square Minimum Discrepancy divided by Degrees of Freedom (CMIN/DF), where acceptable fit indicated by values < 5.0 [25,26]. The significance of the indirect effect was tested using 5,000 bootstrap bias-corrected 95% confidence intervals (CI), with a 95% CI not including zero indicating significance, and a p – value of < 0.05 (two-tailed) was considered statistically significant.

## Ethical considerations

This study was reviewed and approved by the Bahir Dar University College of Education Institutional Research Review Committee (IRRC) (Approval No. 001954, dated 23/09/2024). As the participants were minors, written informed consent was obtained from their parents/guardians, and assent was obtained from the students themselves. Prior to participation, both students and their parents/guardians were provided with a comprehensive information sheet outlining the study's purpose, procedures, potential benefits, and the voluntary nature of participation. They were also informed of their right to withdraw from the study at any stage without penalty.

All participants were treated with fairness, respect, and impartiality throughout the study. To ensure data protection, all information collected was stored securely in password-protected files accessible only to the research team. Identifiable details were removed during analysis to preserve participant anonymity in all disseminated findings. Furthermore, ongoing monitoring of data handling practices was undertaken to safeguard confidentiality and maintain adherence to ethical research standards.

## Results

### Descriptive statistics

The descriptive statistics for 483 participants including 197 males and 286 females, highlight significant gender differences in engagement constructs. As presented in Table 1, females reported higher mean scores in academic task value compared to males. Similarly, females exhibited greater mean scores in student-teacher relationships than males. In terms of cognitive academic task engagement, females again outperformed males. Behavioral academic task engagement also favored females, as did affective academic task engagement. Finally, in positive development outcomes, females had a higher mean compared to males.

Independent samples t-tests confirmed that these mean differences were statistically significant. Specifically, females scored significantly higher than males on academic task value ($t$ (481) = –5.44, $p < 0.001$), student–teacher relationship ($t$ (481) = –3.64, $p < 0.001$), cognitive engagement ($t$ (481) = –2.35, $p = 0.019$), behavioral engagement ($t$ (481) = –4.13, $p < 0.001$), affective engagement ($t$ (481) = –2.90, $p = 0.004$), and positive development outcomes ($t$ (481) = –4.62, $p < 0.001$) (see Table 2).

**Table 1. Descriptive statistics (mean and standard deviation).**

| Variables | Gender | Mean | Std. Deviation |
|---|---|---|---|
| (1) Academic task value (ATV) | All (483) | 3.02 | 0.65 |
| | Male (197) | 2.83 | 0.67 |
| | Female (286) | 3.15 | 0.60 |
| (2) Student-teacher relationship (STR) | All (483) | 3.07 | 0.62 |
| | Male (197) | 2.95 | 0.64 |
| | Female (286) | 3.16 | 0.59 |
| (3) Cognitive engagement (COG) | All (483) | 2.88 | 0.60 |
| | Male (197) | 2.80 | 0.59 |
| | Female (286) | 2.93 | 0.60 |
| (4) Behavioral engagement (BEH) | All (483) | 2.97 | 0.68 |
| | Male (197) | 2.82 | 0.72 |
| | Female (286) | 3.08 | 0.63 |
| (5) Affective engagement (AFF) | All (483) | 2.95 | 0.58 |
| | Male (197) | 2.86 | 0.60 |
| | Female (286) | 3.01 | 0.56 |
| (6) Positive development (PD) | All (483) | 2.92 | 0.44 |
| | Male (197) | 2.81 | 0.42 |
| | Female (286) | 2.99 | 0.44 |

**Table 2. Independent samples t-test results for gender differences in academic task value, student–teacher relationship, task engagement dimensions, and positive development.**

| | Levene's Test for Equality of Variances | | t-test for Equality of Means | | | | | 95% Confidence Interval of the Difference | |
|---|---|---|---|---|---|---|---|---|---|
| | F | Sig. | t | df | Sig. (2-tailed) | Mean Difference | Std. Error Difference | Lower | Upper |
| ATV | .963 | 0.327 | −5.438 | 481 | 0.000 | −0.317 | 0.058 | −0.43213 | −0.20273 |
| STR | 2.668 | 0.103 | −3.636 | 481 | 0.000 | −0.206 | 0.057 | −0.31715 | −0.09462 |
| COG | 0.106 | 0.745 | −2.345 | 481 | 0.019 | −0.130 | 0.055 | −0.23852 | −0.02105 |
| BEH | 5.603 | 0.018 | −4.129 | 481 | 0.000 | −0.255 | 0.062 | −0.37655 | −0.13372 |
| AFF | 3.088 | 0.080 | −2.899 | 481 | 0.004 | −0.155 | 0.053 | −0.25966 | −0.04987 |
| PD | 1.062 | 0.303 | −4.624 | 481 | 0.000 | −0.185 | 0.040 | −0.26372 | −0.10642 |

ATV: academic task value; STR: student – teacher relationship; COG: cognitive engagement; BEH: behavioral engagement; AFF: affective engagement; and PD: positive development.

## Correlation for full, male, and female sample

The bivariate Pearson's correlation analysis reveals significant relationships among student-teacher relationships, perceived academic task value, cognitive academic task engagement, behavioral academic task engagement, affective academic task engagement, and positive development outcomes (see Table 3).

In the overall sample, strong positive correlations exist between student-teacher relationships and academic task value, cognitive engagement, behavioral engagement, affective engagement, and positive development outcomes. Perceived academic task value correlates positively with various engagement types and positive outcomes. Gender-specific

**Table 3. Correlation for full, male, and female samples.**

| Variables | Sample | (1) | (2) | (3) | (4) | (5) | (6) |
|---|---|---|---|---|---|---|---|
| (1) Student-teacher relationship (STV) | Full | 1 | | | | | |
| | Male | 1 | | | | | |
| | Female | 1 | | | | | |
| (2) Academic task value (ATV) | Full | 0.545** | 1 | | | | |
| | Male | 0.540** | 1 | | | | |
| | Female | 0.518** | 1 | | | | |
| (3) Cognitive engagement (COG) | Full | 0.524** | 0.529** | 1 | | | |
| | Male | 0.541** | 0.572** | 1 | | | |
| | Female | 0.499** | 0.485** | 1 | | | |
| (4) Behavioral engagement (BEH) | Full | 0.533** | 0.436** | 0.493** | 1 | | |
| | Male | 0.504** | 0.356** | 0.487** | 1 | | |
| | Female | 0.532** | 0.458** | 0.484** | 1 | | |
| (5) Affective engagement (AFF) | Full | 0.338** | 0.341** | 0.382** | 0.437** | 1 | |
| | Male | 0.354** | 0.324** | 0.381** | 0.387** | 1 | |
| | Female | 0.298** | 0.319** | 0.368** | 0.455** | 1 | |
| (6) Positive development (PD) | Full | 0.621** | 0.563** | 0.548** | 0.619** | 0.453** | 1 |
| | Male | 0.596** | 0.548** | 0.557** | 0.576** | 0.437** | 1 |
| | Female | 0.619** | 0.538** | 0.531** | 0.630** | 0.442** | 1 |

** Correlation is significant at the 0.01 level (2-tailed).

analyses show that males exhibit slightly stronger correlations, particularly between student-teacher relationships and perceived academic task value.

## Analysis of gender differences

**Measurement invariance.** The researchers employed measurement invariance before computing multi - group structural equation modeling (MGSEM) for gender difference. The goodness-of-fit of models were evaluated using an extensive set of indices from various families endorsed by the specialized literature including a "relative chi-square" test, which is the chi-square to the degree of freedom ($X^2/df$), Goodness-of-Fit Index (GFI), Comparative Fit Index (CFI), Tucker-Lewis index (TLI), Root Mean Square Error of Approximation (RMSEA), and Standardized Root Mean Square Residual (SRMR) [25,26]. Goodness-of-fit was decided based on the cut-off criteria expanded in the literature: SRMR < 0.08, RMSEA < 0.08, incremental indices (GFI/CFI/TLI) > 0.90, and a CMIN/df ratio < 5.0 are indicative of an adequate model fit [25,27,28]. As it is presented in Table 4, the model fit of configural, metric, and scalar level measurement invariance model were good, and the measurement invariance of all constructs was reached at scaler level (strong level), which is the necessary condition to conduct multi-group structural equation modeling for gender difference [29].

**Multi - group structural equation modeling (MGSEM) for gender differences.** The results indicate that the student–teacher relationship and students' perceived academic task value were associated with cognitive engagement, explaining 40.3% of the variance among male students and 31.9% among female students. Regarding affective engagement, these factors were associated with 15.0% of the variance for males and 12.6% for females. In terms of behavioral engagement, the model was associated with 26.4% of the variance for males and 32.8% for females. Furthermore, these predictors were associated with 52.3% of the variance in positive development outcomes for male students and 54.9% for female students (see Table 5).

**Table 4. Measurement invariance model fit tests of configural, metrics, and scaler.**

| Variables | Model | Model Fit Tests | | | | | Decision |
|---|---|---|---|---|---|---|---|
| | | X2/df | CFI | TLI | RMSEA | SRMR | |
| Student-teacher relationship | Configural | 2.117 | 0.91 | 0.90 | 0.048 | 0.0641 | Accepted |
| | Metrics | 2.081 | 0.91 | 0.90 | 0.047 | 0.0727 | Accepted |
| | Scaler | 2.075 | 0.91 | 0.90 | 0.047 | 0.0769 | Accepted |
| Academic task value | Configural | 3.269 | 0.97 | 0.93 | 0.069 | 0.0193 | Accepted |
| | Metrics | 2.527 | 0.97 | 0.95 | 0.056 | 0.0250 | Accepted |
| | Scaler | 2.573 | 0.97 | 0.95 | 0.057 | 0.0309 | Accepted |
| Academic task engagement | Configural | 1.568 | 0.96 | 0.95 | 0.034 | 0.0575 | Accepted |
| | Metrics | 1.506 | 0.96 | 0.96 | 0.032 | 0.0536 | Accepted |
| | Scaler | 1.537 | 0.96 | 0.95 | 0.033 | 0.0564 | Accepted |
| Positive development | Configural | 1.545 | 0.91 | 0.90 | 0.034 | 0.0659 | Accepted |
| | Metrics | 1.550 | 0.91 | 0.90 | 0.034 | 0.0669 | Accepted |
| | Scaler | 1.546 | 0.91 | 0.90 | 0.034 | 0.0734 | Accepted |

**Table 5. Squared multiple correlations ($R^2$) for male and female.**

| Variables | Male | Female |
|---|---|---|
| | $R^2$ | $R^2$ |
| Cognitive engagement | 0.403 | 0.319 |
| Affective engagement | 0.150 | 0.126 |
| Behavioral engagement | 0.264 | 0.328 |
| Positive development | 0.523 | 0.549 |

**Direct effects.** As it presented in Table 6, the multi-group structural equation modeling (MGSEM) analysis reveals significant gender differences across various pathways, highlighting nuanced interactions among constructs related to academic engagement. In examining the relationship between student-teacher relationships and behavioral academic task engagement, males demonstrate a strong positive path coefficient ($\beta = 0.440$, B = 0.491, $p < 0.001$), while females exhibit a slightly lower yet significant coefficient ($\beta = 0.403$, B = 0.431, $p < 0.001$). This finding indicates a marginally stronger effect of student – teacher relationship on behavioral academic task engagement for males.

In terms of affective academic task engagement, males benefit significantly from student – teacher relationship, with a path coefficient of ($\beta = 0.253$, B = 0.237, $p = 0.001$), whereas females show a lower but still significant effect ($\beta = 0.181$, B = 0.172, $p = 0.005$). Conversely, the relationship between perceived academic task value and behavioral academic task engagement is non-significant for males ($\beta = 0.119$, B = 0.127, $p = 0.103$), but it is significant and stronger for females ($\beta = 0.249$, B = 0.261, $p < 0.001$) (see Table 6). This suggests that academic task value plays a more critical role in enhancing behavioral engagement among females.

Both genders exhibit significant positive relationships between student – teacher relationship and cognitive academic task engagement, with males at ($\beta = 0.327$, B = 0.301, $p < 0.001$) and females at ($\beta = 0.338$, B = 0.346, $p < 0.001$). Additionally, the path from perceived academic task value to cognitive academic task engagement is significant for both genders, with males at ($\beta = 0.395$, B = 0.349, $p < 0.001$) and females at ($\beta = 0.309$, B = 0.309, $p < 0.001$) (see Table 5), suggesting a stronger impact of perceived academic task value on cognitive academic task engagement for males.

As it presented in Table 6, the relationship between affective academic task engagement and perceived academic task value reveals that males exhibit a coefficient of ($\beta = 0.187$, B = 0.168, $p = 0.017$), while females show a stronger effect at

**Table 6. Multi - group structural equation modeling (MGSEM) for direct relationship.**

| Variables | | | Male | | | | | Female | | | | |
|---|---|---|---|---|---|---|---|---|---|---|---|---|
| | | | **B** | **S.E.** | **C.R.** | **β** | **P** | **B** | **S.E.** | **C.R.** | **β** | **P** |
| BEH | <— | STR | 0.491 | 0.081 | 6.040 | 0.440 | *** | 0.431 | 0.061 | 7.093 | 0.403 | *** |
| AFF | <— | STR | 0.237 | 0.073 | 3.230 | 0.253 | 0.001 | 0.172 | 0.061 | 2.803 | 0.181 | 0.005 |
| BEH | <— | ATV | 0.127 | 0.078 | 1.630 | 0.119 | 0.103 | 0.261 | 0.059 | 4.391 | 0.249 | *** |
| COG | <— | STR | 0.301 | 0.060 | 4.994 | 0.327 | *** | 0.346 | 0.058 | 5.922 | 0.338 | *** |
| COG | <— | ATV | 0.349 | 0.058 | 6.029 | 0.395 | *** | 0.309 | 0.057 | 5.412 | 0.309 | *** |
| AFF | <— | ATV | 0.168 | 0.071 | 2.389 | 0.187 | 0.017 | 0.209 | 0.060 | 3.482 | 0.225 | *** |
| PD | <— | BEH | 0.159 | 0.033 | 4.796 | 0.276 | *** | 0.200 | 0.033 | 6.017 | 0.292 | *** |
| PD | <— | AFF | 0.093 | 0.037 | 2.534 | 0.136 | 0.011 | 0.103 | 0.033 | 3.132 | 0.133 | 0.002 |
| PD | <— | COG | 0.093 | 0.045 | 2.078 | 0.133 | 0.038 | 0.092 | 0.035 | 2.665 | 0.128 | 0.008 |
| PD | <— | STR | 0.149 | 0.044 | 3.370 | 0.231 | *** | 0.212 | 0.039 | 5.389 | 0.289 | *** |
| PD | <— | ATV | 0.133 | 0.040 | 3.295 | 0.214 | *** | 0.115 | 0.037 | 3.119 | 0.160 | 0.002 |

ATV: academic task value; STR: student – teacher relationship; COG: cognitive engagement; BEH: behavioral engagement; AFF: affective engagement; PD: positive development; *B*: unstandardized coefficient beta; *S.E.*: standard error; *β*: standardized coefficient beta, *C.R.*: Critical Ratio; *P*: value of significance level; and *** is $p < 0.001$.

(β = 0.225, B = 0.209, p < 0.001). Furthermore, both genders demonstrate significant effects of behavioral academic task engagement on positive development outcomes, with males at (β = 0.276, B = 0.159, p < 0.001) and females at (β = 0.292, B = 0.200, p < 0.001), indicating a stronger effect for females.

Lastly, cognitive academic engagement significantly influences positive development for both genders, with males at (β = 0.133, B = 0.093, p = 0.038) and females at (β = 0.128, B = 0.092, p = 0.008), indicating a stronger effect for females. The path from student – teacher relationship to positive development outcomes is significant for both genders, with males at (β = 0.231, B = 0.149, p < 0.001) and females at (β = 0.289, B = 0.212, p < 0.001), more pronounced for females compared to males. Additionally, the influence of perceived academic task value on positive development outcomes is more pronounced for males (β = 0.214, B = 0.133, p < 0.001) compared to females (β = 0.160, B = 0.115, p = 0.002) (see Table 6).

**Indirect effects.** As it presented in Table 7, the relationship between student-teacher relationship and positive developmental outcomes, mediated by cognitive academic task engagement, is significant for both male and female students. Specifically, the path coefficients indicate that males exhibit a coefficient of (β = 0.028, 95% CI [0.002, 0.058], p = 0.036), while females show a coefficient of (β = 0.032, 95% CI [0.004, 0.061], p = 0.028). The overlapping confidence intervals suggest comparable effects, though slightly stronger for females. In terms of behavioral academic task engagement, both genders demonstrate statistically significant effects, with males presenting a coefficient of (β = 0.078, 95% CI [0.036, 0.130], p < 0.001) and females a coefficient of (β = 0.086, 95% CI [0.047, 0.132], p < 0.001), indicating a more robust relationship for females. Lastly, regarding affective academic task engagement, both males (β = 0.022, 95% CI [0.002, 0.045], p = 0.024) and females (β = 0.036, 95% CI [0.002, 0.038], p = 0.023) exhibit significant effects, with a slightly more pronounced effect observed in males.

The result of MGSEM revealed that the relationship between students' perceived academic task value and their positive development outcomes, mediated by cognitive academic task engagement shows significant positive effect for both genders. Specifically, males exhibited a path coefficient of (β = 0.33, 95% CI [0.002, 0.070], p = 0.036), while females demonstrated a coefficient of (β = 0.028, 95% CI [0.003, 0.059], p = 0.028), indicating a comparable influence of perceived academic task value through cognitive engagement. In contrast, behavioral academic task engagement showed a non-significant effect for males (β = 0.020, 95% CI [−0.005, 0.054], p = 0.118), while it was significant for females (β = 0.052, 95% CI [0.025, 0.085], p < 0.001), highlighting a stronger pathway for females. Affective academic task engagement

**Table 7. User-defined estimands: (male and female – default model).**

| Parameter | Male | | | | Female | | | |
|---|---|---|---|---|---|---|---|---|
| | Estimate | Lower | Upper | P | Estimate | Lower | Upper | P |
| STR to PD via COG | 0.028 | 0.002 | 0.058 | 0.036 | 0.032 | 0.004 | 0.061 | 0.028 |
| STR to PD via BEH | 0.078 | 0.036 | 0.130 | 0.000 | 0.086 | 0.047 | 0.132 | 0.000 |
| STR to PD via AFF | 0.022 | 0.002 | 0.045 | 0.024 | 0.018 | 0.002 | 0.038 | 0.023 |
| ATV to PD via COG | 0.033 | 0.002 | 0.070 | 0.036 | 0.028 | 0.003 | 0.059 | 0.028 |
| ATV to PD via BEH | 0.020 | −0.005 | 0.054 | 0.118 | 0.052 | 0.025 | 0.085 | 0.000 |
| ATV to PD via AFF | 0.016 | −0.001 | 0.043 | 0.068 | 0.022 | 0.005 | 0.048 | 0.003 |
| Total indirect effect of STR | 0.128 | 0.077 | 0.187 | 0.000 | 0.136 | 0.083 | 0.194 | 0.000 |
| Total indirect effect of ATV | 0.069 | 0.022 | 0.121 | 0.004 | 0.102 | 0.058 | 0.153 | 0.000 |
| Total effect of STR | 0.277 | 0.160 | 0.390 | 0.000 | 0.347 | 0.263 | 0.431 | 0.000 |
| Total effect of ATV | 0.201 | 0.102 | 0.297 | 0.000 | 0.217 | 0.136 | 0.299 | 0.000 |

STR is student – teacher relationship; ATV is academic task value; PD is positive development outcome; COG is cognitive engagement; BEH is behavioral engagement; and AFF is affective engagement.

showed gender-specific effects. For males, the path coefficient was positive but not statistically significant ($\beta = 0.016$, 95% CI [−0.001, 0.043], $p = 0.068$), whereas for females the effect was significant ($\beta = 0.022$, 95% CI [0.005, 0.048], $p = 0.003$) (see Table 7).

## Discussion

The findings of the study emphasize the critical role of classroom contextual factors, particularly student-teacher relationships and perceived academic task value in explaining variations in academic task engagement and positive developmental outcomes, with notable gender differences. This aligns with prior research such as [30], which showed the dynamic interplay between contextual supports and student engagement, emphasizing that both relational and instructional experiences are central to students' academic and developmental success. [31] further elaborated that emotionally supportive teacher-student relationships promote higher levels of behavioral and emotional engagement. Complementing this [32] showed that autonomy-supportive instruction enhances cognitive and behavioral engagement. The significant role of perceived task value, as the study of [8], students are more likely to engage when academic tasks are personally meaningful. This is reinforced by [33], who found that perceived value of academic tasks significantly shapes students' emotional and cognitive engagement, contributing to improved academic and developmental outcomes.

The current study's findings supported with existing research that highlights gender-based differences in how classroom contextual factors influence academic engagement. Females often exhibit stronger connections between perceived task value and engagement. For instance, [34] found that girls were more likely to engage behaviorally and emotionally in academic tasks they perceive as meaningful and aligned with their personal goals. A study by [35] further showed that gender differences in achievement motivation are often linked to varying emotional and cognitive responses to academic tasks. Furthermore, a meta-analysis of school-based interventions targeting student motivational-affective factors found significant positive effects for both genders, with descriptively larger effect sizes for female students, although the difference between genders was not statistically significant [36]. In the Ethiopian context, this stronger link between task value and engagement may be influenced by cultural expectations that encourage diligence, discipline, and compliance in schooling, with education often viewed as a pathway to upward mobility and family contribution for females. These findings suggest that targeted interventions considering these contextual factors could enhance academic engagement and positive development across genders.

The findings also showed that student-teacher relationships were found to directly and positively influence all dimensions of academic task engagement across both genders, with slightly stronger effects for males, particularly in behavioral and affective engagement. This aligns with [31] who highlighted that students' sense of relatedness with teachers enhances their motivation and emotional involvement in learning, with relatedness to teachers being a more salient predictor of engagement for boys. Similarly, [6] conducted a meta-analysis confirming that positive teacher-student relationships lead to higher academic motivation with the effects being stronger in studies with more boys, suggesting that boys may benefit more from relational support. In Ethiopia's educational and cultural environment, males may rely more on teacher relationships because traditional gender norms often grant them greater autonomy, making external relational support from teachers an important source of guidance, structure, and encouragement. Furthermore, perceived academic task value directly predicted higher behavioral engagement for both genders, though more strongly for females. At the same time, perceived task value exerted a stronger influence on cognitive engagement for male students, indicating that boys may rely more on the perceived usefulness and meaning of academic tasks to sustain deeper cognitive investment. This relates with the expectancy-value model [13] which posits that students' achievement-related choices are influenced by their expectations for success and the subjective value they place on the task. [8] Further support this, emphasizing that students, especially girls, are more likely to exert effort and persist when they perceive academic tasks as meaningful and valuable. Recent work by [37] reinforces this, showing that task value is particularly critical for female students in sustaining engagement and academic persistence over time.

The analysis of indirect effects also revealed that gender-specific pathways in classroom context variables influence positive developmental outcomes. Cognitive engagement significantly mediated these effects for both genders, with a slightly stronger influence observed among male students. This finding aligns with research by [32] who emphasized the role of engagement, students' proactive involvement in learning, enhancing motivation and developmental outcomes. Similarly, [38] emphasize the role of community and motivational contexts in positive youth development, highlighting the importance of cognitive engagement. This aligns with findings by [34] who reported that girls were more likely to engage behaviorally and emotionally in academic tasks they perceive as meaningful and aligned with their personal goals. Conversely, behavioral engagement more robustly mediated the relationship between task value and positive development for female students. Also, [8] emphasized that female students generally show greater behavioral and emotional investment in school when task value is perceived as high. Additionally, affective engagement was identified as a significant mediator, particularly for female students, aligning with existing literature that emphasizes how emotional investment in learning environments fosters key personal attributes such as caring, confidence, and connection core components of the Five Cs of PYD [8,31].

The current study supports prior research studies that academic engagement functions as a key mechanism through which classroom experiences foster developmental competencies [8,30]. More importantly, the findings highlight the need for gender-responsive strategies. For boys, enhancing teacher-student relationships may be especially effective in strengthening affective and cognitive engagement, thereby supporting personal growth, a view supported by [6,31] who emphasize the importance of relational support for male students' motivation. This is particularly relevant in Ethiopia, where boys are often socialized toward independence and may need stronger relational anchoring from teachers to remain engaged. For girls, emphasizing the value and utility of academic content may better promote consistent behavioral engagement and foster broader developmental outcomes, consistent with findings from [34,35] which link girls' achievement motivation to task value and emotional engagement in personally meaningful learning contexts. In recent times, such tendencies have been reinforced by Ethiopian cultural norms that value girls' persistence, diligence, and responsibility in education, making task relevance especially motivating for them. These implications align with calls from scholars for more individualized and developmentally aligned instructional practices that support both academic success and positive youth development [6,31].

## Conclusion and implications

The findings from the Multi-Group Structural Equation Modeling (MGSEM) highlight important gender differences in how student-teacher relationships and perceived academic task value contribute to students' academic engagement and positive development. The model explained a substantial proportion of the variance in cognitive, affective, and behavioral engagement, as well as in positive development outcomes, for both male and female students with a slightly stronger explanatory power for female students in terms of positive development. Student-teacher relationships emerged as a robust predictor of all three forms of engagement, particularly for male students in behavioral and affective domains. On the other hand, perceived academic task value showed a stronger impact on cognitive engagement for males and on behavioral engagement for females. Notably, both direct and indirect paths indicated that academic engagement serves as a significant mediator between classroom factors and youth developmental outcomes, with affective and behavioral engagement showing stronger mediating effects for females.

These results suggest several practical implications. First, the gendered nature of engagement underscores the need for educators to apply gender-sensitive strategies. For male students, strengthening emotional connections and cognitive stimulation through engaging tasks and positive teacher relationships may enhance their developmental outcomes. For female students, emphasizing the relevance and value of academic tasks may be more effective in promoting behavioral engagement and subsequent development. Second, the consistently strong impact of student-teacher relationships across both genders indicates that relationship-building should be a central component of teacher professional development. Teachers should be equipped with skills to foster trust, responsiveness, and emotional support in the classroom. Third, instructional practices should aim to enhance students' perceptions of academic task value by connecting learning materials to their personal goals and real-world applications, particularly to encourage higher engagement among female students. Furthermore, academic engagement should be seen not only as a means to academic success but also as a developmental asset that supports competence, confidence, character, connection, and caring. These insights also highlight implications for teacher education and school policy. Teacher training programs should integrate modules on fostering gender-responsive pedagogy and relational skills, while school policies should prioritize creating supportive classroom climates that recognize engagement as a pathway to holistic student development.

However, this study has some limitations that should be considered. First, it relied on self-reported measures, which may be subject to social desirability bias or inaccuracies in self-assessment. Second, the cross-sectional design precludes any causal inferences. Third, the sample was somewhat skewed toward female students (59.2% of the final sample), which may affect the generalizability of the findings; however, this proportion is consistent with the actual gender distribution in Bahir Dar secondary schools, suggesting that the sample remains reasonably representative of the student population. Future research could address these limitations by incorporating multiple data sources, such as teacher ratings and observational measures, and by using longitudinal designs to better examine causal relationships. Despite these limitations, the study provides valuable insights into the differentiated roles of teacher-student relationships and perceived academic task value in fostering gender-responsive educational practices that promote academic engagement and holistic student development.

## Supporting information

**S1 Data. Raw data for the analysis.**
(XLSX)

## Acknowledgments

We extend our heartfelt gratitude to the students who participated in this study by completing the questionnaires. Additionally, we express our appreciation to the school principals and teachers who facilitated the data collection process.

## Author contributions

**Conceptualization:** Getahun Tadesse Abren, Reda Darge Negasi, Amare Sahle Abebe.

**Data curation:** Getahun Tadesse Abren.

**Formal analysis:** Getahun Tadesse Abren.

**Investigation:** Getahun Tadesse Abren, Reda Darge Negasi, Amare Sahle Abebe.

**Methodology:** Getahun Tadesse Abren, Reda Darge Negasi, Amare Sahle Abebe.

**Project administration:** Getahun Tadesse Abren.

**Resources:** Getahun Tadesse Abren.

**Software:** Getahun Tadesse Abren.

**Supervision:** Getahun Tadesse Abren, Reda Darge Negasi, Amare Sahle Abebe.

**Validation:** Getahun Tadesse Abren, Reda Darge Negasi, Amare Sahle Abebe.

**Visualization:** Getahun Tadesse Abren, Reda Darge Negasi, Amare Sahle Abebe.

**Writing – original draft:** Getahun Tadesse Abren.

**Writing – review & editing:** Getahun Tadesse Abren, Reda Darge Negasi, Amare Sahle Abebe.

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
