## [Decision Letter · Decision Letter 0]

15 Sep 2025

Dear Dr. Abren,

Thank you for submitting your manuscript to PLOS ONE. After careful consideration, we feel that it has merit but does not fully meet PLOS ONE’s publication criteria as it currently stands. Therefore, we invite you to submit a revised version of the manuscript that addresses the points raised during the review process.

**Thank you for your submission. Please revise your paper, highlight the changes, and provide a response letter.**
**You need to improve the theoretical framework, enrich the discussion, and update the refs.**
**Best,**
**Ali Derakhshan**

We look forward to receiving your revised manuscript.

Kind regards,

Ali Derakhshan

Academic Editor

PLOS ONE

Journal Requirements:

4. In the online submission form you indicate that your data is not available for proprietary reasons and have provided a contact point for accessing this data. Please note that your current contact point is a co-author on this manuscript. According to our Data Policy, the contact point must not be an author on the manuscript and must be an institutional contact, ideally not an individual. Please revise your data statement to a non-author institutional point of contact, such as a data access or ethics committee, and send this to us via return email. Please also include contact information for the third party organization, and please include the full citation of where the data can be found.

Reviewers' comments:

Reviewer's Responses to Questions

**Comments to the Author**

1. Is the manuscript technically sound, and do the data support the conclusions?

Reviewer #1: Yes

Reviewer #2: Yes

2. Has the statistical analysis been performed appropriately and rigorously?

Reviewer #1: Yes

Reviewer #2: Yes

3. Have the authors made all data underlying the findings in their manuscript fully available?

Reviewer #1: No

Reviewer #2: Yes

4. Is the manuscript presented in an intelligible fashion and written in standard English?

Reviewer #1: Yes

Reviewer #2: Yes

Reviewer #1: The study is well-targeted to a crucial and relevant educational psychology problem employing a good methodological strategy to examine gender-differentiated pathways. The findings have helpful implications for educationally gender-sensitive practice. However, several of the points require clarification and elaboration to enhance the contribution and scholarship of the manuscript. Please to see attached Review Report

Reviewer #2: Major Points for Consideration

1. Sampling and Representation:

The sample is heavily skewed towards female students (59.2% final sample). While this can happen in random sampling, it should be briefly acknowledged in the limitations section. A comment on whether this proportion is representative of the general student population in Bahir Dar would be useful.

The average age of participants is provided (16.69), but the age range would be helpful context.

2. Clarity in Results Reporting:

Table 4 (R² values): The p-values for all R² estimates are listed as .001 or .002. However, R² is a measure of variance explained, and its significance is typically inferred from the overall model fit and path significances. Reporting a p-value for R² itself is unusual. Please confirm the statistical test being reported here.

Page 27, Indirect Effects: The text states: "females a coefficient of (β = 0.086, 40% CI [0.036, 0.135])". This appears to be a typo and should read "95% CI".

Page 28, Table 6: The estimates for "Total Indirect Effect of STR" and "Total Indirect Effect of ATV" are listed as 0.000 for both estimate and CI, which seems implausible given the specific indirect effects listed above are not zero. This likely requires correction.

3. Discussion Nuance:

The discussion could briefly speculate on the cultural reasons behind the observed gender differences within the Ethiopian context. Why might teacher relationships be more salient for boys? Why might task value be a stronger driver for girls? Adding a sentence or two would enrich the interpretation.

When citing references in the discussion, ensure they perfectly align with the point being made. For example, on Page 30, the citation [33] is about "school-based collaboration," which seems less directly relevant to the point about motivational interventions than other potential citations might be.

Minor comments

Required Editorial Corrections

1. Abstract:

The beta (β) symbols in the abstract are currently used to represent R² (variance explained). This is statistically incorrect. Beta (β) represents standardized regression coefficients. The abstract should state "R² = 0.403" instead of "β = 0.403". This error appears throughout the abstract. This is a critical correction.

2. Data Availability Statement:

The statement is currently appropriate for restricted data. However, ensure the journal's policy is met. The text "The datasets that support the conclusions of this study are available from the corresponding author upon reasonable request" is standard and acceptable.

3. Formatting and Typos:

Page 8, Line 31-32: 95% CI [0.279, 0.516], p < .001), while they explain 31.9% of the variance for female students (β = 0.319, 95% CI [0.224, 0.408], p < .001). -> Again, these should be R² values, not β.

Page 13, Line 143: Bahir Dar City Administration, which includes a total of 21 secondary schools (– i.e., 11 public and 10 private) -> The format "(– i.e., ...)" is awkward. Use "(i.e., 11 public and 10 private)".

Page 17, Line 239: 5 point Lickert scale -> Should be "5-point Likert scale".

Page 21, Table 1: The last variable is labeled "Positive engagement (PD)" but should be "Positive development (PD)" to match the text and construct.

Page 25, Table 5: The column header C.R. is defined as "Critical ratio" in the note. It is more commonly referred to as the "Critical Ratio" (which is equivalent to a z-score for testing parameter estimates). This is fine, but ensure consistency.

Page 30, Line 462: had significa3nt positive effects -> Typo: "significant".

4. Reference Consistency:

Page 36, Ref 6: Rey Educ Res. 2011; -> Typo: Should be "Rev Educ Res."

Please perform a final thorough check to ensure all in-text citations have a corresponding entry in the reference list and that all references are correctly formatted according to PLOS ONE guidelines.

**Do you want your identity to be public for this peer review?** For information about this choice, including consent withdrawal, please see our Privacy Policy

Reviewer #1: No

Reviewer #2: **Yes: ** Prof. Ekramul Islam

---

## [Author Response · Author response to Decision Letter 1]

30 Sep 2025

Please provide additional details regarding participant consent. In the ethics statement in the Methods and online submission information, please ensure that you have specified (1) whether consent was informed and (2) what type you obtained (for instance, written or verbal, and if verbal, how it was documented and witnessed). If your study included minors, state whether you obtained consent from parents or guardians. If the need for consent was waived by the ethics committee, please include this information.

Responses: - All participants were treated with fairness, respect, and impartiality. Data were stored securely in password-protected files accessible only to the research team, and identifying details were removed to ensure anonymity. Ongoing monitoring was conducted to safeguard confidentiality and uphold ethical standards.

The revised Ethics Statement in the manuscript now reflects these details.

---

## [Decision Letter · Decision Letter 1]

3 Dec 2025

Gender dynamics in the relationship among student-teacher relationships, academic task engagement, academic task value, and positive developmental outcomes

PLOS ONE

Dear Dr. Abren,

Thank you for submitting your revised manuscript to PLOS ONE.  Our two reviewers have completed the review and the results they gave were both positive. Therefore, after making the revisions based on the reviewers' comments, it can be accepted

We look forward to receiving your revised manuscript.

Kind regards,

Xiaopeng Wu

Academic Editor

PLOS ONE

Journal Requirements:

Additional Editor Comments:

This revision has basically addressed the comments raised by the reviewers. After the revision based on the reviewers' opinions, it can be accepted

Reviewers' comments:

Reviewer's Responses to Questions

**Comments to the Author**

Reviewer #1: All comments have been addressed

Reviewer #2: All comments have been addressed

2. Is the manuscript technically sound, and do the data support the conclusions?

Reviewer #1: Yes

Reviewer #2: Yes

3. Has the statistical analysis been performed appropriately and rigorously?

Reviewer #1: Yes

Reviewer #2: Yes

4. Have the authors made all data underlying the findings in their manuscript fully available?

Reviewer #1: Yes

Reviewer #2: Yes

5. Is the manuscript presented in an intelligible fashion and written in standard English?

Reviewer #1: Yes

Reviewer #2: Yes

Reviewer #1: The authors are to be congratulated on a strong piece of research. The revisions have significantly strengthened the manuscript. I recommend acceptance. A final, careful proofread is recommended to polish minor grammatical issues and improve sentence flow.

Reviewer #2: This is a well-structured, methodologically sound, and theoretically grounded study that addresses an important and under-researched topic—gender differences in educational processes and outcomes in an Ethiopian context. The use of Multi-Group Structural Equation Modeling (MGSEM) is appropriate and rigorous. The manuscript is clearly written, and the findings have meaningful implications for educational practice and policy. I recommend acceptance with minor revisions.

Minor Concerns and Suggestions for Revision

1. Abstract

The abstract is clear but could be slightly more accessible to a broader audience. Consider rephrasing statistical results (e.g., “account for” → “were associated with”) to better reflect the correlational design.

2. Introduction

The link between academic task value and positive developmental outcomes could be more explicitly theorized. Consider strengthening the theoretical justification using Expectancy–Value Theory.

3. Methods

Sampling: Clarify how the 13 classrooms were selected (e.g., random lottery from all Grade 9–10 classrooms in the five schools).

Model Fit Indices: The manuscript correctly reports CMIN/DF < 5 as acceptable, but earlier versions contained an error (CMIN/DF < 0.50). Ensure this correction is consistent throughout.

4. Results

Table 1 (Descriptive Statistics): Include results of independent samples t-tests (t-values, p-values) to support claims of gender differences.

R² Reporting: Ensure that all references to R² are correctly labeled (not as β) and that p-values are not incorrectly associated with R².

Confidence Intervals: Correct minor typos (e.g., “40% CI” → “95% CI”) in indirect effects reporting.

5. Discussion

Cultural Context: Briefly speculate on how Ethiopian cultural norms (e.g., gender roles, educational values) may explain the observed gender differences.

Citation Alignment: Ensure all citations directly support the claims made (e.g., replace [33] with a more relevant reference if needed).

6. Limitations

Explicitly acknowledge the cross-sectional design as a limitation for causal inference.

Note the gender imbalance in the sample (59.2% female), but clarify that this reflects the actual student population in Bahir Dar.

7. References

Ensure all references are complete and correctly formatted (e.g., Reference #6 should include full author list: Roorda et al., 2011).

Prefer citing foundational theoretical works (e.g., Lerner et al., 2005, 2011) over measurement tool references when discussing the Five Cs model.

Editorial and Formatting Issues

Correct minor typographical errors (e.g., “Lickert” → “Likert”, “significant” → “significant”).

Ensure all acronyms are defined at first use (e.g., PYD on page 14).

Standardize decimal places and formatting across tables.

**Do you want your identity to be public for this peer review?** For information about this choice, including consent withdrawal, please see our Privacy Policy

Reviewer #1: No

Reviewer #2: **Yes: ** Prof. Dr. Md. Ekramul Islam

---

## [Author Response · Author response to Decision Letter 2]

11 Dec 2025

Thank you for the opportunity to revise our manuscript. We have carefully reviewed and addressed all comments provided by the academic editor and reviewers. A detailed, point-by-point response is included in the attached document, and all corresponding revisions have been made in the revised manuscript. We appreciate the constructive feedback, which has significantly improved the clarity, methodological transparency, theoretical grounding, and overall quality of the paper.

---

## [Editor Report · Decision Letter 2]

16 Dec 2025

Gender dynamics in the relationship among student-teacher relationships, academic task engagement, academic task value, and positive developmental outcomes

PONE-D-25-33636R2

Dear Dr Getahun Tadesse Abren, M.A.

We’re pleased to inform you that your manuscript has been judged scientifically suitable for publication and will be formally accepted for publication once it meets all outstanding technical requirements.

Kind regards,

Xiaopeng Wu

Academic Editor

PLOS One

---

## [Editor Report · Acceptance letter]

PONE-D-25-33636R2

PLOS One

Dear Dr. Abren,

I'm pleased to inform you that your manuscript has been deemed suitable for publication in PLOS One. Congratulations! Your manuscript is now being handed over to our production team.

Kind regards,

on behalf of

Professor Xiaopeng Wu

Academic Editor

PLOS One